# The Neoangiogenic Transcriptomic Signature Impacts Hepatocellular Carcinoma Prognosis and Can Be Triggered by Transarterial Chemoembolization Treatment

**DOI:** 10.3390/cancers16203549

**Published:** 2024-10-21

**Authors:** Rosina Maria Critelli, Federico Casari, Alberto Borghi, Grazia Serino, Cristian Caporali, Paolo Magistri, Annarita Pecchi, Endrit Shahini, Fabiola Milosa, Lorenza Di Marco, Alessandra Pivetti, Simone Lasagni, Filippo Schepis, Nicola De Maria, Francesco Dituri, María Luz Martínez-Chantar, Fabrizio Di Benedetto, Gianluigi Giannelli, Erica Villa

**Affiliations:** 1Gastroenterology Unit, CHIMOMO Department, University of Modena and Reggio Emilia, 41124 Modena, Italy; rosinamaria.critelli@unimore.it (R.M.C.); fabiola.milosa@unimore.it (F.M.); pivetti.alessandra@aou.mo.it (A.P.); simone.lasagni@unimore.it (S.L.); demaria.nicola@aou.mo.it (N.D.M.); 2Radiology, Azienda Ospedaliero-Universitaria di Modena, University of Modena and Reggio Emilia, 41125 Modena, Italy; casari.federico@aou.mo.it (F.C.); caporali.cristian@aou.mo.it (C.C.); annarita.pecchi@unimore.it (A.P.); 3Internal Medicine, Ospedale di Faenza, 48018 Faenza, Italy; elbarto5@tin.it; 4National Institute of Gastroenterology “IRCCS Saverio de Bellis”, Research Hospital, 70013 Castellana Grotte, Italy; grazia.serino@irccsdebellis.it (G.S.); endrit.shahini@irccsdebellis.it (E.S.); francesco.dituri@irccsdebellis.it (F.D.); gianluigi.giannelli@irccsdebellis.it (G.G.); 5HPB Surgery and Liver Transplant Unit, Azienda Ospedaliero-Universitaria di Modena, University of Modena and Reggio Emilia, 41125 Modena, Italy; 6Clinical and Experimental Medicine PhD Program, 41125 Modena, Italy; lor.dimarco@gmail.com; 7M.E.C. Dipartimental Unit, University of Modena and Reggio Emilia, 41125 Modena, Italy; filippo.schepis@unimore.it; 8Liver Disease Laboratory, Centre for Cooperative Research in Biosciences (CIC bioGUNE), Basque Research and Technology Alliance (BRTA), Bizkaia Technology Park, Building 801A, 48160 Derio, Spain; mlmartinez@cicbiogune.es; 9Centro de Investigacion Biomedica en Red de Enfermedades Hepaticas y Digestivas (CIBERehd), 28200 Madrid, Spain

**Keywords:** neoangiogenic transcriptomic signature, hypoxia, microRNA expression, survival

## Abstract

Therapeutic response and survival outcomes in hepatocellular carcinoma (HCC) patients remain unsatisfactory, with a 5-year survival rate of less than 25%. The lack of molecular analysis of HCC tissue hinders the identification of precise predictors of disease progression and treatment outcomes. Analyzing the hepatic neoangiogenic transcriptomic signature can predict the biological aggressiveness of HCC and its resistance to therapies, offering a valuable diagnostic and prognostic tool that can significantly enhance the management of HCC.

## 1. Introduction

Hepatocellular carcinoma (HCC) remains a global health challenge despite advances in screening and treatment, causing over 700,000 deaths annually with a five-year survival rate under 10% [1]. HCC commonly arises as a complication in patients with long-term liver cirrhosis, affecting roughly a third of this population [2]. Prognosis hinges on the stage of liver disease, tumor traits, and treatment [1].

Until now, prognostic evaluations and treatment decisions have been based on staging systems like the widely adopted Barcelona Clinic Liver Cancer (BCLC) classification [3], supported by European (EASL) [4] and American (AASLD) guidelines [5]. However, BCLC’s predictive accuracy at the individual level has been questioned. The main drawbacks include imprecise stratification of patients into the BCLC-B subclass and a certain degree of rigidity in stage-specific therapeutic choices, leading to poor adherence to its therapeutic indications in real-life clinical practice. As a result, its use in everyday clinical practice has become more limited [6,7,8,9]. Furthermore, most staging systems, including BCLC, consider only initial tumor characteristics, overlooking tumor growth and the emergence of new lesions [10].

There is a growing consensus that integrating clinical staging with biological markers of the tumor could refine prognoses and guide treatment strategies [11,12]. Nonetheless, the complexity and limited added value of many identified molecular signatures have hindered their adoption in clinical settings. Additionally, most signatures are based on retrospective data from a small fraction of HCC patients eligible for resection, limiting their applicability [13].

One tumor characteristic linked to prognosis is the growth rate [14,15,16]. Fast-growing HCCs, constituting 20–30% of cases, are notably aggressive and less responsive to treatments [15,16,17]. In our previous work in 2016 [15], we identified a five-gene neoangiogenic transcriptomic signature (nTS) through an extensive microarray study. This signature includes angiopoietin-2, Delta-like canonical notch ligand 4, neuropilin and Tolloid-like 2, endothelial cell-specific molecule 1, and nuclear receptor subfamily 4 group A member 1, all associated with neoangiogenesis. HCCs in patients with the nTS exhibited not only an extremely fast growth rate but also a distinctly immunosuppressed microenvironment, evidenced by the local upregulation of PD1 (Programmed cell death protein 1) and PD-L1 (Programmed Death Ligand 1) [18]. Additionally, these HCCs showed prominent epithelial–mesenchymal transition and clear activation of TGFβ1 signaling. Overall, the nTS was linked with aggressive HCCs and poor prognosis [18]. In this study, we aim to evaluate how this signature and its changes after treatment affect patient outcomes and survival in two prospectively enrolled HCC cohorts.

## 2. Patients and Methods

### 2.1. Patients

From January 2010 to January 2017, the Gastroenterology Unit at Azienda Ospedaliero-Universitaria, Modena, prospectively enrolled newly diagnosed treatment-naïve HCC patients detected via biannual ultrasound surveillance. Treatment followed global guidelines: early-stage HCCs underwent surgical resection or radiofrequency ablation (RFA), intermediate stages received transarterial chemoembolization (TACE), and advanced stages were treated with systemic therapies like sorafenib. Upon sorafenib cessation, options included regorafenib or participation in clinical trials, such as milciclib (*n* = 2) or nivolumab (*n* = 2). Best supportive care (BSC) was reserved for those with advanced liver disease or poor performance status (PS) [Child–Pugh (CP) B 8/9 and/or PS > 1]. Liver transplant eligibility and MELD score adjustments followed Italian transplant guidelines [19]. Tumor response was measured according to the modified RECIST criteria [20]. Response was defined after six months of treatment as stable disease (SD) or partial response (PR) or complete response (CR). Post-treatment imaging with contrast-enhanced Computed Tomography or Magnetic Resonance Imaging was conducted at one month and every three months for two years post complete treatment. Recurrences triggered stage-appropriate retreatment. Those without recurrence for over a year were monitored with semi-annual ultrasounds.

We confirmed our findings with a validation cohort, also prospectively enrolled, in our unit from 2018–2021, due to the unavailability of external cohorts with suitable baseline biopsies for HCC. This group underwent identical diagnostic procedures to the derivation cohort.

Conducted in line with the Declaration of Helsinki and clinical trial practices, the protocol was approved by an ethics committee, with all participants providing informed consent (IRB10/08_CE_UniRer; ClinicalTrials ID: NCT01657695).

### 2.2. Neoangiogenic Transcriptomic Signature

After written informed consent, all patients underwent an ultrasound-guided biopsy for both histological examination and transcriptomic analysis to determine the presence of the neoangiogenic transcriptomic signature (nTS) [15]. In patients experiencing recurrence, after collection of informed consent, a second biopsy or a third biopsy was obtained. In these cases, histological examination and transcriptomic analysis were also performed. Tumor and non-tumor tissue were collected in ice-cold RNA-later and processed within 24 h ([15] and Appendix A). In instances of repeated treatments, another biopsy was performed to assess any changes in the nTS. The physicians responsible for patient care were blinded to the nTS results, ensuring that this information did not influence the therapeutic decision-making process.

### 2.3. microRNA Analysis

We utilized a segment of the ultrasound-guided biopsy for miRNA expression analysis. This was performed using the miRCURY LNA RT Kit from Qiagen s.r.l. (Milan, Italy) ([18] and Appendix A). The following miRNAs were evaluated: miR-221-3p and miR-222-3p (involved in promotion of proliferation, migration, invasion, and hypoxia-driven angiogenesis) [21,22,23]; mir-15b-5p and mir-16-5p (inhibition of apoptosis, growth, and upregulation in the hypoxic environment) [24]; miR-30a-5p (inhibition of proliferation, invasion, tumor growth) [25,26,27]; miR-30d-5p (inhibition of autophagy) [28,29]; miR-145-5p (negative regulation of cell proliferation) [30,31]; mir-122-5p (cell cycle arrest, EMT, and apoptosis) [32,33]; and miR-210 (hypoxia) [34,35,36,37].

### 2.4. Statistical and Bioinformatic Analysis

Continuous variables were presented as means ± standard deviations (SDs), and comparisons were made using Student’s *t*-test. Categorical variables were expressed as frequencies of patients with and without the nTS and were compared using Pearson’s χ^2^ test or Fisher’s exact test, depending on group size. The Mann–Whitney U test was employed for comparisons when data were assumed to be non-normally distributed. Treatments were categorized as curative, endoarterial, or systemic, and analyzed across up to six treatment courses. The data did not report missing values; therefore, a complete case analysis was performed.

Patients were censored at liver transplantation (LT), death, or last follow-up. Missing death dates were sourced from hometown registries.

Survival probabilities were calculated using the Kaplan–Meier method, with the log-rank test comparing different treatments. The primary survival analysis was by type of treatment group and the secondary survival analysis was by presence or absence of nTS.

Univariable and multivariable Cox proportional hazards regression analyses were used to identify factors associated with patients’ survival. Child–Pugh and MELD scores were not included to avoid collinearity. Baseline variables collected included age, sex, performance status, etiology, bilirubin, albumin, International Normalized Ratio, creatinine, ascites, encephalopathy, number of HCC nodules (one, two, three, or multiple), presence of portal vein thrombosis, and transcriptomic signature [15]. Five competing-risk Cox proportional hazards regression models were developed to account for the importance of different collinear factors. All statistical tests were performed with a two-sided significance threshold set at *p* < 0.05.

miRNA targets were predicted by means of three tools as miRWalk (http://mirwalk.umm.uni-heidelberg.de/) [38], miRDB (http://www.mirdb.org/) [39], and miRabel (http://bioinfo.univ-rouen.fr/mirabel/) [40].

PASW Statistics (ver. 28; IBM Corporation, Armonk, NY, USA) was used for statistical analysis.

## 3. Results

### 3.1. Clinical Results

Three hundred and twenty-eight patients were enrolled in this prospective study. Median observation time was 31 months (mean ± SD: 40.5 ± 35.2 months). Patients enrolled in this study were mostly males (*n* = 259, 79.0%). The mean age at diagnosis was 65.1 ± 11.2 (median 67) years. Viral etiology was prevalent (64.9%; of these, 53.9% HCV-positive and 11.0% HBV-positive). Alcohol abuse was the only etiologic factor in 16.5% of subjects and was associated with HCV in 9.1% of patients. NASH was present in 18.3% of cases. Liver function was preserved in most patients (69.2% in Child–Pugh Class A, MELD score of 10.5 ± 3.8, median 9.0). nTS+ patients more often had multifocal presentation (nTS+ vs. nTS−: 39.4% vs. 18.3%; *p* < 0.001), were more often Edmondson–Steiner grade 3 or 4 (74.1% vs. 57.3%, *p* = 0.028), and had higher AFP levels (3.896 ± 13.265 vs. 954 ± 6.209 ng/mL, *p* = 0.012). Composition of derivation and validation cohorts was similar (Appendix A).

### 3.2. Therapeutic Management and Outcome

At diagnosis, 15.9% (52 patients) received only supportive care, with no significant difference between nTS− and nTS+ groups. nTS+ patients were more likely to need systemic therapy from the onset (26.7% vs. 11.9%, *p* = 0.0012). A total of 276 patients (84.1%) received at least one HCC treatment, with 484 treatments administered overall (Appendix A). The median intervals between the first, second, and third treatments were 30, 33, and 22 weeks, respectively.

Notably, fewer nTS+ patients proceeded beyond the second treatment. The average number of treatments was similar between nTS− and nTS+ groups. As initial treatments, resection and RFA were common, but nTS+ patients were significantly less likely to be suitable for RFA (*p* = 0.003). TACE was also more frequent in nTS− patients as a first treatment (*p* = 0.01).

In subsequent treatments, the use of curative methods decreased, while endoarterial treatments remained high. Liver transplants were more common in later treatments but were still rare overall (*n* = 42, 8.6%). Among these, only 13.6% were nTS+ patients, often ineligible for transplant due to advanced disease or rapid progression.

Three months post-treatment, around 36% of patients achieved an objective response, significantly more among nTS− compared to nTS+ patients (*p* = 0.0042). The cure rate was low, around 20%, with no significant difference between nTS− and nTS+ groups. These rates declined to 18.0% for responses and 8.5% for cures after the second treatment. Few patients progressed to a third treatment, with both response and cure rates diminishing further. Appendix A displays outcomes of sequential treatments, showing proportions of patients achieving cure receiving non-curative treatments like TACE, systemic therapy, or BSC. Appendix A illustrate the outcomes for nTS− and nTS+ patients, emphasizing the significant differences in cure rates and mortality between the groups (*p* < 0.001).

### 3.3. Survival Analysis

#### 3.3.1. Kaplan–Meier Analysis

By this study’s end, 79.9% of participants had died, with a median survival of 31 months (Appendix A). Kaplan–Meier analysis revealed that nTS+ patients had a significantly shorter survival of 13 months, compared to 41 months for nTS− patients (Appendix A, *p* < 0.0001). This trend was consistent with that of the validation cohort (Appendix A).

Twenty-six patients (7.9%) died after an average of 16 months (median 12 months) before they could receive a second treatment. The mortality rate following the first treatment was significantly higher in the nTS+ group compared to the nTS− group (33.7% vs. 16.5%; *p* = 0.004). Survival was significantly better for patients receiving any treatment, with a median survival of 44 months, compared to those receiving only BSC or systemic therapy, with median survivals of 7 and 11 months, respectively. Patients undergoing multiple consecutive treatments had a median survival of 57 months (*p* < 0.0001, log-rank test) (Figure 1A). Survival disparities were pronounced between nTS− and nTS+ patients across all treatment categories, with nTS+ patients experiencing poorer outcomes. Median survivals were 9 vs. 6 months for BSC, 19 vs. 9 months for systemic therapy, 47 vs. 20 months for at least one treatment, and 68 vs. 33 months for multiple treatments for nTS− vs. nTS+ patients (*p* < 0.0001, log-rank test) (Figure 1C,D,E,F), despite similar rates of multiple treatments between nTS+ and nTS− patients (24.4% vs. 29.8%; *p* = 0.202, Fisher’s exact test). The validation cohort confirmed these patterns, showing consistent treatment responses and survival rates for the whole cohort (Figure 1B) and in the treatment subgroups (Figure 1G,H,I,J).

Overall survival across the entire cohort varied significantly depending on the type of dominant treatment administered, defined as the treatment with the highest potential therapeutic impact. LT yielded the longest median survival at 123 months, outpacing other treatments like RFA at 42 months, surgical resection at 52 months, TACE at 34 months, systemic therapy at 11 months, and BSC at 6 months (*p* < 0.001) (Figure 2A,B). For nTS− patients, survival benefits scaled up from supportive to more curative treatments. Their median survival times varied by treatment type: 123 months for LT, 76 months for resection, 40 months for TACE, 44 months for RFA, 17 months for systemic therapy, and 9 months for BSC. Conversely, nTS+ patients experienced limited benefits across the treatment spectrum, with liver transplant outcomes being less favorable (median survival of 34 months) compared to resection (33 months), RFA (52 months), TACE (29 months), systemic therapy (9 months), and BSC (5 months). A direct comparison between each type of treatment is shown in Figure 2C–H for the derivation cohort and in Figure 2I–N for the validation cohort.

#### 3.3.2. Cox Regression Analysis

We conducted a univariate Cox regression to assess the impact of factors like transcriptomic signature, performance status, liver transplant, BCLC stage, albumin, CRP, nodule count, and portal vein thrombosis on survival, all showing significant associations (Table 1). In multivariate analysis, we adjusted for collinearity, excluding CRP and creating five models that included either transcriptomic signature, portal vein thrombosis, or nodule number. The transcriptomic signature and the nodule number emerged as independent survival predictors. A liver transplant was a consistent positive predictor, and higher albumin levels also correlated with better survival in two models. Both the transcriptomic signature’s negative impact and the liver transplant’s positive impact were confirmed in the validation cohort.

### 3.4. Effect of Repeat Treatments on Transcriptomic Signature and microRNA

In a cohort of 105 TACE patients, 86 underwent the procedure twice, and 43 three times; for RFA, 26 out of 59 required a second treatment. Biopsies to assess transcriptomic shifts were performed on 80 TACE patients before their second treatment and on all patients before their second RFA treatment. After the first TACE, 21 patients (26.2%) shifted from nTS− to nTS+ status. This shift increased to 58.1% (25 of 43) after the second TACE, compared to only 7.7% (2 of 26) in RFA patients, showing a statistically significant difference between the treatments (*p* = 0.039). This transition in TACE patients significantly affected survival, with a median survival of 25 months for those who transitioned to nTS+ status, as opposed to 35 months for those who did not (*p* = 0.030). This became apparent 18 months post-treatment (Figure 3). 

The transition involved changes in tumor microRNA expression levels related to angiogenesis, proliferation, cell cycle, and hypoxia. Initial microRNA levels were similar between transarterial chemoembolization (TACE) and radiofrequency ablation (RFA) patients, but significant changes occurred after treatment, especially post-TACE. Most alterations were seen after the second TACE biopsy, while minimal changes followed RFA treatment. Notably, TACE-related changes were mostly unfavorable, whereas RFA-induced alterations, such as increased levels of miR-145-5p or miR-30a-5p, were neutral or beneficial (Figure 4).

### 3.5. microRNA–mRNA Integrative Analysis

We performed a bioinformatic analysis to locate potential targets of the miRNAs within a five-gene hepatic signature, which includes ANGPT2, DLL4, NETO2, ESM1, and NR4A1. The analysis revealed significant post-TACE changes in miR-221-3p, miR-222-3p, and miR-210-3p, which are implicated in the regulation of five key genes (Table 2). This regulatory activity suggests that these miRNAs can play a role in modulating the hepatic transcriptomic landscape.

## 4. Discussion

Recent advances in therapy have improved outcomes for hepatocellular carcinoma (HCC) patients, yet prognosis remains generally poor [2]. Patient treatment is largely determined by prognostic scoring systems like the Barcelona Clinic Liver Cancer (BCLC) classification [3]. These systems, while empirically validated, often fail to consider individual patient differences and the diverse biological behavior of HCC [6,7,8]. Critically, these scores do not account for the intrinsic biological characteristics of the tumor, which can profoundly influence therapeutic efficacy and the interpretation of clinical outcomes, including the likelihood of recurrence and patient mortality. In this study, we did not use the nTS [15] for therapeutic allocation. Instead, we meticulously assessed each patient’s tumor at diagnosis and, when feasible, after treatment. This approach allowed us to analyze the effects of treatment against the backdrop of the tumor’s initial and changing biological characteristics. Data from the derivation cohort, confirmed by the validation cohort (despite the latter having been matured later than the derivation cohort and during later therapeutic times), revealed that the transcriptomic signature divides the HCC cohort into two distinct groups: nTS+ and nTS−. The nTS+ group, characterized by rapid tumor growth and specific molecular markers, had limited access to curative treatments like RFA or transplantation. This limitation correlated with worse disease progression and higher mortality rates; alarmingly, 57% of nTS+ patients died from HCC after initial treatment, with the proportion rising to 70% after a second treatment. This rapid deterioration could be due to the nTS+ tumors’ tendency to be multifocal at diagnosis, making them ineligible for curative treatments. Besides multifocality, nTS+ HCCs show high PD1/PD-L1 expression, significant epithelial–mesenchymal transition, and an immunosuppressive microenvironment with systemic inflammation [18]. These features suggest that immunotherapy may be a beneficial therapeutic option for this subgroup, as demonstrated in patients with an immunosuppressed microenvironment, evaluated using a radiomic non-invasive score [41].

The conventional treatment algorithm, based on general criteria [3,4,5], may exclude the nTS+ group from effective treatments. The fast and dynamic changes in their tumor burden, due to high growth rates, make these patients unsuitable for curative interventions. Notably, surgical resection and RFA yielded the most favorable outcomes within the nTS+ subgroup, presumably due to reduced wait times for treatment. Conversely, liver transplantation was infrequent among nTS+ patients, attributable to stringent eligibility criteria, with these patients demonstrating suboptimal survival outcomes. In stark contrast, nTS− patients benefited substantially from LT, outperforming other treatment modalities.

These findings advocate for the incorporation of the transcriptomic signature in the pre-transplant evaluation to enhance the therapeutic benefit. Lee et al. [42] emphasized the judicious utilization of liver transplants, considering the organs’ shortage, to maximize patient outcomes. The integration of tumor biological markers into existing allocation algorithms may attenuate HCC recurrence and extend patient survival. This perspective is reinforced by Duvoux et al. [43], who evidenced the prognostic enrichment afforded by alpha-fetoprotein (AFP) within the Milan criteria [44]. Similarly, the Metroticket 2.0 model [45], which combines conventional metrics with AFP levels, surpasses its AFP-excluded counterparts in prognostic accuracy. However, this model’s efficacy is tempered by its reliance on static baseline factors and its partial capture of biological aggression as indicated by AFP levels alone [15]. In contrast, our transcriptomic signature, with its correlation to tumor growth kinetics, transcends these conventional scores in prognostic precision. A comparative analysis of liver transplant recipients demonstrated that endothelial angiopoietin-2, the histological correlate of the transcriptomic signature, surpassed all clinical scores in recurrence prediction, with a concurrent independent association with patient survival [46]. Furthermore, Receiver Operating Characteristic curve analysis corroborated the superior specificity and sensitivity of this molecular marker over other scores, including Metroticket_AFP [46].

Collectively, these insights underscore that while LT confers a survival advantage for nTS− HCCs, the nTS+ subgroup necessitates alternative strategic considerations beyond resection and RFA. The distinctive molecular profile of nTS+ patients, characterized by PD1–PDL1 overexpression and active neoangiogenesis [18], points to the promise of contemporary systemic therapies that coalesce immune checkpoint inhibitors with antiangiogenic agents. These novel therapeutic approaches have exhibited efficacy superior to conventional treatments, including LT. In an nTS+ cohort with post-transplant HCC recurrence unresponsive to sorafenib, the nivolumab and bevacizumab combination resulted in a significantly extended median overall survival compared to second-line tyrosine kinase inhibitor therapies [47].

Our study’s longitudinal biopsy protocol after treatment initiation revealed a critical observation: the transformation of liver tissue biology following TACE intervention. Over half of patients exhibited an aggressive tumor signature after the initial TACE sessions, a shift not observed with RFA. This change was further confirmed by different miRNA expression patterns post-treatment. Although baseline miRNA levels were consistent across TACE and RFA cohorts, post-treatment analyses unveiled significant elevations in miR-145-5p, miR-221-3p, miR-222-3p, and miR-210-3p following TACE. After TACE, miR-145-5p, a suppressor of cell proliferation and invasion [31], was downregulated, whereas miR-221-3p, miR-222-3p, and miR-210-3p were upregulated. These miRNAs promote liver carcinogenesis by targeting the CDK inhibitor p27 and enhancing cell growth in vitro [23,31,48]. MiR-210-3p is implicated in liver carcinogenesis and tumor angiogenesis through targets like SMAD4 and STAT6 [34] and is driven by hypoxia-inducible factor 1α [35]. This upregulation might worsen hypoxia in HCC, potentially increasing carcinogenesis [36,37,49]. In contrast, RFA favorably modulated these miRNAs, particularly increasing miR-145-5p and decreasing miR-210-3p. In addition to TACE-related hypoxia and miR-210-3p upregulation, miR-221-3p and miR-222-3p increase may also contribute to HCC progression and therapy resistance [34,35]. The significant changes post-TACE suggest these miRNAs might drive the shift to a more aggressive tumor state. Intriguingly, our in silico analysis has revealed that all five genes in the transcriptomic signature could be influenced by these miRNAs. This finding suggests a potential mechanistic role for miRNAs in the transition from bland to aggressive HCC phenotypes, which is characterized by a marked increase in angiogenesis [15]. Not surprisingly, very recent data from the Emerald-1 study, which compared TACE alone, TACE with durvalumab, and TACE combined with both durvalumab and bevacizumab, have shown a significant improvement in outcomes, specifically in the arm treated with bevacizumab [50]. This suggests that the addition of a specific VEGF inhibitor can substantially improve the efficacy of TACE by targeting the VEGF increase and angiogenesis induced by the treatment [51,52]. The recognition of the potentially relevant role of TACE-induced hypoxia has also led to the study of combining TACE with Tirapazamine, a prodrug whose anticancer effects are activated under low-oxygen conditions and which reverts to an inactive form when exposed to oxygen [53]. This approach has yielded promising results in preclinical [54,55] and early clinical trials [56,57].

The authors acknowledge several limitations of this study. Firstly, it was conducted at a single center, which may affect the generalizability of the results. Secondly, the small size of the biopsy samples limited our ability to explore a wide array of molecular mechanisms. Consequently, this restriction also narrowed the number of microRNAs we could examine. We therefore selectively tested microRNAs involved in angiogenesis, proliferation, cell cycle, and hypoxia. These pathways were chosen because they are highly relevant to the observed events and critical for understanding the underlying mechanisms. Furthermore, the invasiveness of the biopsy procedure limited the collection to only one follow-up biopsy, or two in some cases, which restricted the exploration of potential additional changes. A longer follow-up period would also have been valuable for identifying other biological or clinical events. Additionally, it has been challenging to find a consistent collection of liver biopsies taken before and after loco-regional treatments. Lastly, but importantly, we have no data on the mechanisms underlying the onset of the nTS, such as whether genetic mutations may contribute to its development. This element will have to be studied as soon as sufficient liver tissue is available for analysis. Despite these limitations, the authors believe that such biopsies could offer new insights and avenues for improving the management of HCC treatment.

## 5. Conclusions

In conclusion, while liver transplantation benefits nTS− HCC patients, the nTS+ group needs different strategic considerations beyond resection and RFA. The unique molecular profile of nTS+ patients, especially PD1–PDL1 overexpression and active neoangiogenesis, suggests the potential of new systemic therapies combining immune checkpoint inhibitors with antiangiogenic agents. These novel combinations have shown greater efficacy than conventional treatments, including LT, in nTS+ patients with post-transplant HCC recurrence unresponsive to sorafenib [46]. Recognizing the dynamic nature of tumor biology, especially following TACE, is crucial as it negatively impacts prognosis. Therefore, a careful approach to treatment selection is warranted, with further research needed on the effectiveness of combined therapies [47,50,53,54,55,56,57].

## Figures and Tables

**Figure 1 cancers-16-03549-f001:**
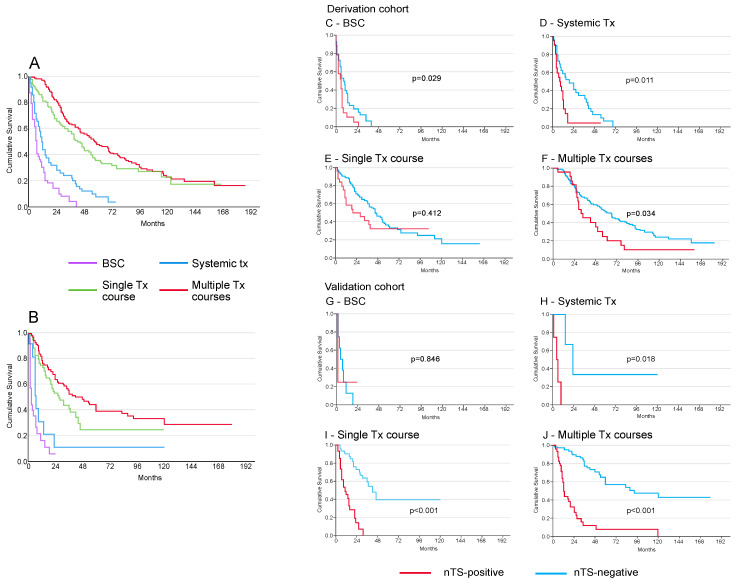
Outcome of treatments (BSC, systemic therapy, at least one treatment, and multiple treatments) according to presence or absence of the transcriptomic signature (nTS) in HCC. The outcomes for the derivation and validation cohorts as a whole are depicted in (**A**,**B**), respectively. Survival for best supportive care (BSC), systemic therapy, one therapeutic course, or multiple therapeutic courses, stratified for presence or absence of nTS, are depicted in (**C**–**F**) (derivation cohort) and (**G**–**J**) (validation cohort) (log-rank test).

**Figure 2 cancers-16-03549-f002:**
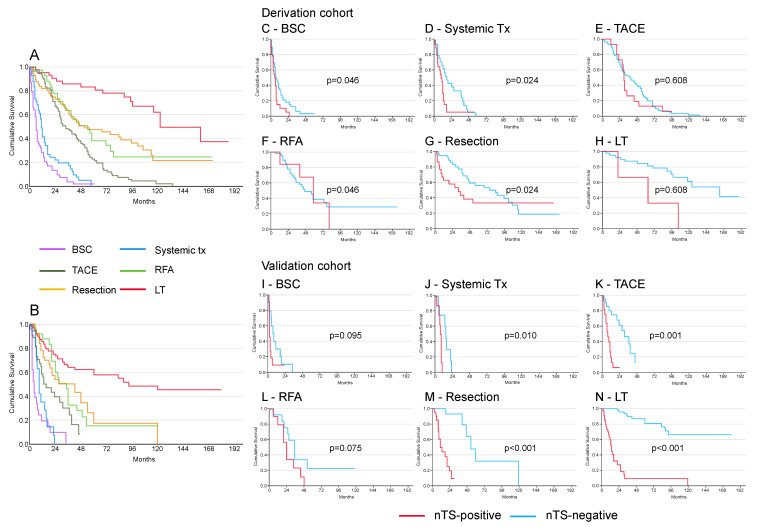
Outcome of treatments according to nTS status. Considering the outcome of all aggregate treatments performed in each patient, a clear-cut survival difference was present depending on the presence, in the totality of treatments performed, of a dominant treatment ((**A**), derivation cohort and (**B**), validation cohort). Stratification of the cohort by transcriptomic signature showed optimal results for LT and progressively worse results for less curative treatments in the nTS− patients ((**C**–**H**), derivation cohort and (**I**–**N**), validation cohort) (*p* < 0.001, log-rank test). Survival was much worse in nTS+ patients, although the difference among treatments was often significant (BCS: best supportive care; Systemic Tx: systemic treatment; TACE: transarterial chemoembolization; RFA: radiofrequency ablation; LT: liver transplantation).

**Figure 3 cancers-16-03549-f003:**
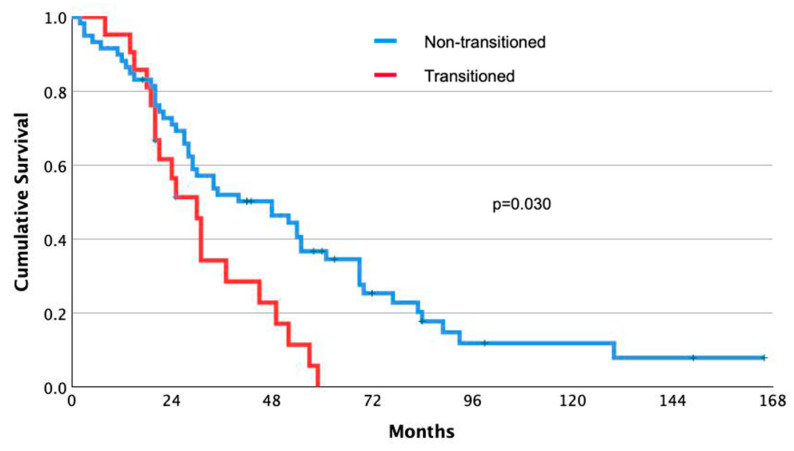
Kaplan–Meier analysis of the survival of HCC patients who received TACE as their first treatment. Repeat biopsies were obtained from 80 of the 86 patients who underwent a second TACE, for comparison with baseline. A significant worsening in median survival was observed with the transition from nTS− to nTS+ status (*p* = 0.030, log-rank test).

**Figure 4 cancers-16-03549-f004:**
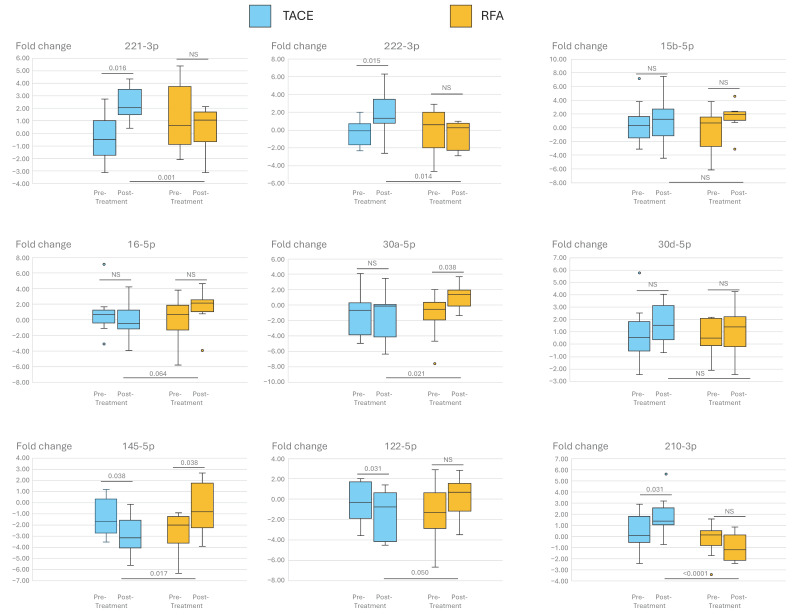
Changes in microRNA expression after TACE (light blue) and RFA (orange) treatments. The left box indicates the baseline fold-change level, and the right box shows the levels after the procedure. Initially, there were no significant differences in microRNA levels between TACE and RFA patients. Post-TACE, notable changes included increased levels of miR-221-3p, miR-222-3p, and miR-210-3p, and decreased levels of miR-145-5p and miR-122-5p. For RFA patients, only miR-30a-5p and miR-145-5p showed increased post-procedure levels. Statistical significance was assessed using paired sample *t*-tests, as detailed in the figure (NS = Non-significant).

**Table 1 cancers-16-03549-t001:** The results from both univariate and multivariable Cox regression analyses, tested using the Wald method, highlight the significant impact of various factors on patient survival. The neoangiogenic transcriptomic signature (nTS) showed collinearity with several baseline factors, such as multifocal tumors, portal vein thrombosis (PVT), Edmondson–Steiner grading, AFP levels, and C-reactive protein (CRP) levels. To mitigate this, five distinct multivariable models were crafted: Model 1 for nTS, Model 2 for tumor multifocality, Model 3 for PVT, Model 4 for Edmondson–Steiner grading, and Model 5 for AFP levels. CRP was omitted from these models due to its collinearity with all variables but the Child–Pugh score, ensuring the accuracy of the prognostic evaluations for these factors on patient outcomes.

	Univariable			Multivariable		
	HR	95% CI	*p*	HR	95% CI	*p*
**Model 1**						
Transcriptomic signature	2.660	1.873–3.790	<0.001	2.444	1.491–4.007	<0.001
Sex	1.136	0.831–1.553	0.425			
Age at diagnosis	1.014	1.002–1.026	0.027	1.009	0.985–1.034	0.466
Performance status	2.010	1.086–3.718	0.026	1.140	0.580–1.816	0.703
LT vs. others	0.180	0.104–0.310	<0.001	0.130	0.055–0.303	<0.001
Surgical resection vs. others	0.448	0.280–1.020	0.001	0.588	0.337–1.025	0.061
Etiology (viral vs. nonviral)	1.248	0.963–1.618	0.094			
Ascites	1.317	0.987–1.75	0.061			
Encephalopathy	0.779	0.249–2.440	0.668			
BCLC stage	1.482	1.140–1.926	0.003	1.215	0.824–1.972	0.275
Bilirubin	0.990	0.935–1.059	0.872			
Albumin	0.787	0.626–0.989	0.040	0.690	0.472–1.032	0.071
Creatinine	1.475	0.938–2.320	0.093			
CRP	1.094	1.031–1.161	0.003			
INR	0.903	0.547–1.490	0.689			
Multiple nodules at presentation	1.305	1.160–1.469	<0.001			
PVT	1.676	1.185–2.371	0.004			
**Model 2**						
Multiple nodules at presentation	1.305	1.160–1.469	<0.001	1.346	1.136–1.565	<0.001
Performance status	2.010	1.086–3.718	0.026	0.974	0.535–2.081	0.877
Age at diagnosis	1.014	1.002–1.026	0.027	1.055	0.993–1.042	0.580
LT vs. others	0.180	0.104–0.310	<0.001	0.123	0.053–0.286	<0.001
Surgical resection vs. others	2.230	1.04–3.566	0.001	0.776	0.446–1.333	0.354
BCLC stage	1.482	1.140–1.926	0.003	1.094	0.796–1.504	0.580
Albumin	0.787	0.626–0.989	0.040	0.486	0.340–0.695	<0.001
**Model 3**						
Portal vein thrombosis	1.676	1.185–2.371	0.004	1.927	0.879–4.223	0.102
Performance status	2.010	1.086–3.718	0.026	0.867	0.411–1.830	0.709
Age at diagnosis	1.014	1.002–1.026	0.027	1.055	0.993–1.042	0.580
LT vs. others	0.180	0.104–0.310	<0.001	0.089	0.037–0.212	<0.001
Surgical resection vs. others	2.230	1.04–3.566	0.001	1.344	0.780–2.31	0.287
BCLC stage	1.482	1.140–1.926	0.003	1.095	0.801–1.496	0.569
Albumin	0.787	0.626–0.989	0.040	0.538	0.378–0.765	<0.001
**Model 4**						
Edmondson–Steiner grading	1.431	1.232–1.662	<0.001	1.555	0.178–2.052	0.002
Performance status	2.010	1.086–3.718	0.026	1.469	0.760–2.841	0.253
Age at diagnosis	1.014	1.002–1.026	0.027	1.006	0.983–1.030	0.614
LT vs. others	0.180	0.104–0.310	<0.001	0.141	0.064–0.310	<0.001
Surgical resection vs. others	2.230	1.04–3.566	0.001	1.380	0.803–2.369	0.244
BCLC stage	1.482	1.140–1.926	0.003	1.301	0.978–1.729	0.070
Albumin	0.787	0.626–0.989	0.040	0.477	0.334–0.680	<0.001
**Model 5**						
AFP levels (median)	1.784	1.328–2.397	<0.001	1.440	0.930–2.230	0.102
Performance status	2.010	1.086–3.718	0.026	1.353	0.669–2.738	0.400
Age at diagnosis	1.014	1.002–1.026	0.027	1.003	0.979–1.028	0.812
LT vs. others	0.180	0.104–0.310	<0.001	0.123	0.053–0.286	<0.001
Surgical resection vs. others	2.230	1.04–3.566	0.001	1.344	0.780–2.31	0.287
BCLC stage	1.482	1.140–1.926	0.003	1.095	0.801–1.496	0.569
Albumin	0.787	0.626–0.989	0.040	0.538	0.378–0.765	<0.001

**Table 2 cancers-16-03549-t002:** Prediction of putative binding sites of analyzed miRNA in the five-gene transcriptomic hepatic signature was performed using three bioinformatic tools: miRWalk [38], miRDB [39], and miRabel [40].

	ANGPT2	DLL4	NETO2	NR4A1	ESM1
miR-221-3p	√	√	√	√	√
miR-222-3p	√	√	√	√	√
miR-15b-5p	√	√		√	
miR-16-5p		√		√	√
miR-30a-5p		√	√		√
miR-30d-5p		√			
miR-145-5p	√	√	√	√	
miR-122-5p	√				
miR-210-3p	√	√	√	√	√

## Data Availability

The data presented in this study are available on request from the corresponding author. The data are not publicly available due to privacy reasons.

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
