# Peer review of "The Neoangiogenic Transcriptomic Signature Impacts Hepatocellular Carcinoma Prognosis and Can Be Triggered by Transarterial Chemoembolization Treatment"

_cancers, 2024, doi:10.3390/cancers16203549_

Round 1
Reviewer 1 Report
Comments and Suggestions for Authors
Overall evaluation:
This study evaluated the relationship between the neoangiogenic transcriptomic signature (nTS) and the clinical symptoms, treatment outcomes, and survival rates of hepatocellular carcinoma (HCC) patients. It prospectively followed 328 patients in a derivation cohort and 256 patients in a validation cohort, collecting and analyzing clinical data, treatment approaches, histological examinations, and transcriptomic analyses. nTS significantly influences treatment eligibility and survival, especially after repeated TACE procedures, altering oncogenic microRNA expression. The study design is sound, innovative, and holds clinical applicability and persuasive value.
Recommendations:
Introduction:
In line 56, there may be an error for the description of treatment1?
Results:
1. In line 156, there may be an error for the description of nTS :39.4%?
2. In Supplementary Table 1, the corresponding p-values are indicated within the table.
3. Please give exact microRNA names related to angiogenesis, proliferation, cell cycle, and hypoxia for better readability.
Discussion:
In this section, the authors mentioned that nTS+ HCCs show high PD1/PD-L1 expression, significant epithelial-mesenchymal transition, and an immunosuppressive microenvironment with systemic inflammation. However, this conclusion is not directly supported by the data in the current study. It is possible that the authors are referencing findings from previous research. If so, the relevant references should be provided to support these claims.
In line 423, there may be an error for the description of neoangiogenesis18?
Author Response
Introduction:
Q - In line 56, there may be an error for the description of treatment1?
A – I apologize for the clumsiness in typing; numbering has been corrected accordingly (line 56)
Results:
Q - 1. In line 156, there may be an error for the description of nTS :39.4%?
A - I apologize for not having caught the error, due to the transition from the text in Word for Mac to the website. I have corrected the typo (now line 174 ).
Q - 2. In Supplementary Table 1, the corresponding p-values are indicated within the table.
A - As suggested, the p-values have been indicated within the table.
Q - 3. Please give exact microRNA names related to angiogenesis, proliferation, cell cycle, and hypoxia for better readability.
A - The exact attribution of the micro RNAs had been indicated in the Supplemental Methods, page 7. We have detailed it in the text (lines 130-136) and we have added the appropriate references (ref. 21-37).
Discussion:
Q - In this section, the authors mentioned that nTS+ HCCs show high PD1/PD-L1 expression, significant epithelial-mesenchymal transition, and an immunosuppressive microenvironment with systemic inflammation. However, this conclusion is not directly supported by the data in the current study. It is possible that the authors are referencing findings from previous research. If so, the relevant references should be provided to support these claims.
A – We apologize for having skipped the citation in the Discussion. Indeed, these data are part of a previous published work, that was extensively cited in the Introduction (lines 79-86, ref 18). We have not re-evaluated these features in the present study, but we have proof that they are constant features in the nTS+ HCC, e.g. the TGFβ1 signaling activation was already evidenced in the original Gut paper (ref. 15) the PD1 increase in the study reporting HCC occurring after DAAs (Faillaci et al., doi: 10.1002/hep.29911). We have now completed the sentence in the Discussion with the appropriate reference (line 393).
Q - In line 423, there may be an error for the description of neoangiogenesis18?
A – I apologize again for the clumsiness in typing; numbering has been corrected accordingly (line 393).
Reviewer 2 Report
Comments and Suggestions for Authors
The article investigates the impact of the neoangiogenic transcriptomic signature (nTS) on treatment outcomes and survival in hepatocellular carcinoma (HCC) patients. It reveals that nTS+ patients have poorer survival rates and limited treatment options, with nTS status changing after repeated TACE treatments, which also alters oncogenic microRNA expression.
1. The methods section should provide more detail on the criteria for patient selection and the specific protocols followed during biopsy and treatment.
2. There should be a clearer description of the statistical tests used and the rationale behind choosing them. Also, the handling of missing data should be explained.
3. Ensure consistent use of terms throughout the manuscript. For instance, ensure 'nTS' is consistently defined when first used.
4. The manuscript would benefit from clearer figures and tables, with better labeling and explan of the data presented.
5. The authors mentioned the potential of immunotherapy, but a more in-depth discussion on the mechanisms and applicable patient populations could be included. For instance, what are the indications, potential side effects, and how can immunotherapy be combined with existing treatments to enhance efficacy? More clinical trial data could also be cited to support these points. Relative reference such as PMID: 38223688 etc could be cited in introduction or discussion. (For example, line 390-392)
6. The discussion on the limitations of the study is brief. It should be expanded to provide a more comprehensive understanding of how these limitations might affect the results and conclusions. For example, it could suggest including long-term follow-up data to better understand the long-term effects of different treatment strategies, especially regarding post-transplant recurrence, changes in microRNA expression, and the long-term efficacy of immunotherapy.
Comments on the Quality of English LanguageMinor editing of English language required.
Author Response
Q - 1. The methods section should provide more detail on the criteria for patient selection and the specific protocols followed during biopsy and treatment.
A – I have integrated the method section as requested (lines 97-102; 118-120).
Q - 2. There should be a clearer description of the statistical tests used and the rationale behind choosing them. Also, the handling of missing data should be explained.
A – I have integrated the method section into the statistical part as requested (lines 138-145; 152-159).
Q - 3. Ensure consistent use of terms throughout the manuscript. For instance, ensure 'nTS' is consistently defined when first used.
A – I have verified throughout the manuscript as indicated.
- The manuscript would benefit from clearer figures and tables, with better labeling and explan of the data presented.
A – Following your request and that of reviewer #3, we have substantially modified Figure 1 and 2 and the tables (including supplementary Table 1).
Q - 5. The authors mentioned the potential of immunotherapy, but a more in-depth discussion on the mechanisms and applicable patient populations could be included. For instance, what are the indications, potential side effects, and how can immunotherapy be combined with existing treatments to enhance efficacy? More clinical trial data could also be cited to support these points. Relative reference such as PMID: 38223688 etc could be cited in introduction or discussion. (For example, line 390-392)
A – I totally agree with reviewer #2 that immunotherapy deserves great attention in the management of HCC patients. Unfortunately, immunotherapy has only recently gained a place in HCC management and only few patients in our cohort could benefit from such an option. On the other hand, it is true that the specific characteristics of nTS+ patients (who constantly have increased PD1/PDl1 expression) could constitute a subgroup who could better benefit from immunotherapy. We have added a comment in the Discussion (lines 359-361) and a reference (reference 41).
- The discussion on the limitations of the study is brief. It should be expanded to provide a more comprehensive understanding of how these limitations might affect the results and conclusions. For example, it could suggest including long-term follow-up data to better understand the long-term effects of different treatment strategies, especially regarding post-transplant recurrence, changes in microRNA expression, and the long-term efficacy of immunotherapy.
A – I agree with Reviewer #2 that extending the follow-up period would be ideal. However, it is important to emphasize that nTS+ patients had significantly shorter survival compared to nTS-negative patients, and the median follow-up was already much longer than their median survival, regardless of nTS status. Nevertheless, we have addressed this limitation more thoroughly in a revised comment within the Discussion (lines 435-443).
Comments on the Quality of English Language
Minor editing of English language required.
We have revised the English language
Reviewer 3 Report
Comments and Suggestions for Authors
This manuscript explores the correlation between a neoangiogenic transcriptomic signature and HCC prognosis. The authors show that analyzing the hepatic neoangiogenic transcriptomic signature can predict the biological aggressiveness of HCC and its resistance to therapies, offering a valuable diagnostic and prognostic tool that can significantly enhance the management of HCC. The study is well designed, and the inferences drawn are well supported by data. However, there are some minor concerns outlined below:
1. Replot Figure 1 showing nTS+ and nTS- in the same graph. Plot different graphs for different treatment courses. This will make it easier to understand differences between nTS+ and nTs- for the same treatment course.
2. Same as above for Figure 2.
3. Line 224: are these median survival for nTS- patients?
4. A major limitation of this study is the absence of genetic data. Is there a correlation between specific mutations present in the patients and the resulting nTS signature? Do specific mutations drive the nTS- to nTS+ transition after TACE treatment? These are very pertinent questions that remain unanswered. If genetic data for these patients is available, it should be utilized to analyze these correlations. If it is not available, this should be discussed in the limitations.
5. Figure 4, X- and Y-axis labelling missing.
6. Data showing high PD1/PD-L1 expression, significant epithelial-mesenchymal transition, and an immuno-suppressive microenvironment with systemic inflammation in nTS+ patients should be provided.
Comments on the Quality of English Language1. Minor grammatical and spelling errors in manuscript that needs correction, example nTS- sometimes written as nTS@.
Author Response
Q - 1. Replot Figure 1 showing nTS+ and nTS- in the same graph. Plot different graphs for different treatment courses. This will make it easier to understand differences between nTS+ and nTs- for the same treatment course.
Q - 2. Same as above for Figure 2.
A – We have redesigned both Figures and Table
Q - 3. Line 224: are these median survival for nTS- patients?
A – We have rewritten the text to reflect the modifications made to Figure 2, which should now provide greater clarity.
Q - 4. A major limitation of this study is the absence of genetic data. Is there a correlation between specific mutations present in the patients and the resulting nTS signature? Do specific mutations drive the nTS- to nTS+ transition after TACE treatment? These are very pertinent questions that remain unanswered. If genetic data for these patients is available, it should be utilized to analyze these correlations. If it is not available, this should be discussed in the limitations.
A – The evaluation of genetic mutations is one of the projects we intended to pursue, but it has not been completed due to the lack of sufficient liver tissue (most data obtained so far have come from biopsy samples). To date, we have been unable to identify any clinical or biological features that could be relevant to the development of nTS. This limitation has been acknowledged in the Discussion (lines 439-442).
Q - 5. Figure 4, X- and Y-axis labelling missing.
A – I added the required labelling for the Y axis (i.e. fold change) and the specification for the x axis (pre and post treatment).
Q - 6. Data showing high PD1/PD-L1 expression, significant epithelial-mesenchymal transition, and an immuno-suppressive microenvironment with systemic inflammation in nTS+ patients should be provided.
A - We have not evaluated these features in the present study. However, we have previously shown that PD1/PD-L1 expression, significant epithelial-mesenchymal transition, and an immuno-suppressive microenvironment is a constant feature of nTS+ HCC patients (Gut paper (ref. 15 and Faillaci et al., doi: 10.1002/hep.29911). This has been delineated in the Introduction lines 79-85).
Comments on the Quality of English Language
Q - 1. Minor grammatical and spelling errors in manuscript that needs correction, example nTS- sometimes written as nTS@.
A - I apologize for the typos, due to the transition from Mac environment to the Cancers website. I have revised the whole manuscript, and I hope that it will not happen again.
Round 2
Reviewer 2 Report
Comments and Suggestions for Authors
I have no other suggestions.
Comments on the Quality of English LanguageMinor editing of English language required.